# A New Method for Identifying the Central Business Districts with Nighttime Light Radiance and Angular Effects

**Na Jie** [1]**, Xin Cao** [1,2,***], Jin Chen** [1,2] **and Xuehong Chen** [1,2]

1   State Key Laboratory of Remote Sensing Science, Faculty of Geographical Science, Beijing Normal University, Beijing 100875, China
2   Beijing Engineering Research Center for Global Land Remote Sensing Products, Faculty of Geographical Science, Beijing Normal University, Beijing 100875, China
*   Correspondence: caoxin@bnu.edu.cn

**Abstract:** Central business districts (CBDs) play a crucial role in urban economic activities. Thus, the location and boundaries of CBDs identified by the unified standards are essential for comparative analyses in urban geography. However, past research mainly applied specific data or sensitive methods to delimitate CBDs within local knowledge in the case study, there remains no automated standardization technique for identifying and delimitating CBDs across the globe. This paper proposed a new method for identifying CBDs based on nighttime lights (NTL) to overcome the above limitations. The main advantages of this method include (1) the use of available high-quality global Black Marble products, which are the basis of a standardized delineation of CBDs and (2) the use of more characteristics of CBD (i.e., the brightness) and NTL negative angular effects that can reflect high-rise building. The proposed method was employed in 14 cities in China and the U.S., and the results showed that China cities needed five NTL indexes and U.S. cities needed two NTL indexes to distinguish CBD and non-CBD successfully. Therefore, our approach is recommended for CBD detection and delineation over large areas.

**Keywords:** Central Business District; nighttime light radiance; NASA Black Marble; localized contour tree; angular effect; urban structure type





## 1. Introduction

The central business district (CBD), a concentration of skyscrapers in the city center, is an increasingly important area in a metropolis [1]. It is a powerful economic engine in the city and a barometer of the city's economy [2]. As the heart of the city, the CBD concentrates a large number of financial, commercial, cultural, and service institutions, and it contains a highly developed transportation network and attracts a bunch of employees and customers [1,3]. The development of the CBD has proven to be a vital contributor to the intensification of local agglomeration economies and business activities, thus making the whole city benefit from the high centrality of commercialization [2,4,5]. Since the CBD is such a small but vital urban spatial structural type, urban planners and scientists often need the location and extent of the CBD to conduct various analyses, such as on-street parking demand [6], land-use patterns [7], housing prices [8,9], commute mode choice [10], and urban growth [11]. Consequently, the importance of the CBD locations and spatial extent is indisputable.

However, the CBD is a vague geographical entity that can be described in words but is hard to depict the area in a complicated urban environment [12,13]. Hence, twentieth-century studies have attempted to identify and delineate the CBD using functional and morphological features such as building height, land use, land value, the contrast between day and nighttime population, shop rent, and volume of sales [14–16]. Among these methods, Murphey-Vance's technique has been widely used because it is more quantitative and easier to get research material [17–19]. Murphy and Vance [20] collected building

height and each floor's land use in downtown by detailed field mapping, and then they proposed central business height index (CBHI) and central business intensity index (CBII) based on these collected data to delineate the CBD. However, this approach's time and labor cost are so high that it is challenging to quickly carry out globally. Remote sensing (RS) and geographic information system (GIS) technologies, therefore, play a key role in the mapping and display of the CBD in efficiency.

In the recent related work, the methods of the CBD detection are divided into two types: GIS-based methods and RS-based methods. In the first type of method based on GIS data, the CBDs are extracted by kernel density estimation (KDE) of the road network and point of interest (POI) data that can reflect socio-economic attributes and human activities [21–25]. For example, Borruso [21] and Wu [22] collected commercial POI datasets in the main center of the study area and used traditional KDE to delimitate CBD. Nevertheless, Yu [23] and Yang [24] think that applying the traditional KDE of the homogeneous hypothesis under Euclidean distance is not reasonable in delimiting urban CBD, considering that socio-economic events are greatly influenced by transportation networks. Accordingly, Yu used network distance rather than Euclidean distance in KDE to identify the CBD [23], and Yang integrated road intersections with KDE to recognize the CBD based on POI [24]. On the contrary, only a small number of remote sensing studies focused on delineating CBD. Through statistical experiments based on 3-D city models of three CBD areas, Taubenboeck et al. [26] validated that morphological and physical parameters (e.g., building height, volume, and floor space density) are important features to distinguish between the CBD and the non-CBD and then performed the classification of the CBDs based on these parameters derived from Cartosat-1 digital surface models. This method achieved the CBD detection and extraction over large areas. Overall, the above works are objective, economical, and fine for the delineation of the CBD in the case study.

However, these approaches also have several limitations. Firstly, although many platforms (e.g., Google Map and Open Street Map) manage POI datasets, there are still problems of incomplete coverage and spatially inconsistent data quality worldwide [27]. Apart from this, it is hard to support global CBD detection with expensive Cartosat-1 imageries [28]. Secondly, KDE is sensitive to bandwidth [29,30] (i.e., different bandwidths generate different density surfaces). Moreover, the optimal bandwidth is inconsistent across cities. For instance, the optimal bandwidths of the two Italian cities are different in Borruso's study [21]. Thirdly, the above GIS-based results are actually fragmented and scattered hot spot spatial patterns caused by KDE itself as a hotspot analysis tool [31]. Researchers pick out the CBD from several hotspots with the help of their local knowledge. In summary, these drawbacks hinder automated standardization techniques that can consistently detect and extract the CBDs in megacities across the globe, which is required for comparative studies of the business districts in the city center [20,26].

Therefore, an available high-quality global dataset, a parameter-insensitive boundary extraction method, and the more straightforward CBD features are the keys to solving the above shortcomings. Recently, Roman et al. [32] applied a sequence of algorithms and auxiliary materials to correct the radiance values based on nighttime lights (NTL) remote sensing imagery acquired in the Day/Night Band (DNB) of the Visible Infrared Imaging Radiometer Suite (VIIRS) and produced the global Black Marble suite. Based on this, NASA released a state-of-the-art NTL product in 2020, the VNP46A2 Daily Moonlight-adjusted NTL Product from January 19, 2012 onward. A wealth of studies [33–37] have shown that VIIRS NTL is widely used in socio-economic parameter estimation (e.g., population, GDP, and urbanization level), urban spatial structure and urban built-up area extraction. In terms of urban spatial structure [38], Chen used a topographic approach and VIIRS NTL intensity to identify the urban spatial structure of Shanghai in 2017. The results of the method show that this method is capable of extracting many different types of precise centers such as main centers, commercial centers, transportation, and industrial centers, and the choice of the threshold value has little effect on the spatial extent of these centers, which is very useful to delimit the CBD under the uniform criteria. However, to be able to recognize

the CBD automatically from urban spatial structures, it is necessary to rely on additional features. Previous researchers have found high nighttime light brightness in CBDs, whereas airports are also very bright [39]. Fortunately, some studies have found angular effects that nighttime light changes with different viewing zenith angles (VZA) [40–43] and determined that blocking light by building with different heights is an essential factor in forming the angular effect [44–47]. Tan et al. [48] explicitly pointed out the angular effect of urban CBDs (i.e., the scatter plot of NTL intensity versus VZA shows a negative linear relationship in CBD areas, while it shows a positive linear relationship or U-shaped curve in other non-CBD areas). In other words, the angular effects of NTL can reflect the magnitude of building height. Therefore, combining Chen's method with the NTL radiance and angular effect is possible to identify the CBDs and extract boundaries under uniform standardization on a global scale based on Black Mable products.

In summary, this paper presents a new method for identifying and delineating the CBD based on nighttime lights. Specifically, this approach relies on Black Marble data to obtain urban spatial structures and angular effects and designs rules that distinguish between CBD and non-CBD urban structures to determine the CBD, according to statistical parameters of annual nighttime lights and negative angular effect pixels in each urban spatial structure. For testing the performance of the proposed method in different countries, experiments are conducted in seven cities in China and the United States, respectively.

## 2. Materials and Methods

### 2.1. Study Area

The concentration of high-rise buildings and commercial activities is a common characteristic of CBDs in large global cities while the urban environment that hosts CBD varies worldwide. Urban spatial structures are formed and affected by population, economics, institution, and history among which national spatial planning and regulations are also important elements in the urban spatial patterns of metropolitan areas [49,50]. Consequently, to examine the performance of the new method in different urban environments, this paper selected China and the United States, both of which have large economies but differ in terms of urbanization processes, political systems, and cultures. Meanwhile, to verify our new method, we chose seven cities in two different countries as study areas: The first requirement was that these urban CBDs are already economically and morphologically well-developed to ensure that they can be detected by NTL remote sensing. The second requirement was that the locations and boundaries of these urban CBDs could be obtained from local urban knowledge or reliable material to ensure that the results of the new method could be validated. According to these requirements, we selected seven cities in China, Beijing, Nanjing, Hangzhou, Shanghai, Chongqing, Guangzhou, and Shenzhen, and seven cities in the USA, Boston, New York, Chicago, Philadelphia, Los Angeles, Dallas, and Houston (Figure 1).

### 2.2. Datasets

#### 2.2.1. NTL Products

In this paper, CBDs were delineated at 15-arc-second resolution from NASA's daily and annual Black Marble products suite (VNP46), which are available from the NASA website (https://blackmarble.gsfc.nasa.gov/). All Black Marble data bands used in this study are listed in Table 1. The daily Black Marble products comprise the daily at-sensor top-of-atmosphere nighttime radiance (VNP46A1) and the daily lunar BRDF-adjusted nighttime lights (VNP46A2). The daily viewing zenith angle (VZA) was obtained from the sensor zenith layer of VNP46A1, and the daily corrected NTL radiance was obtained from the DNB_BRDF-Corrected_NTL layer of VNP46A2. Daily quality flags for cloud contamination reduction were collected from the Mandatory_Quality_Flag layer and QF_Cloud_Mask layer of VNP46A2. Finally, the urban NTL intensity distribution was represented by the AllAngle_Composite_Snow_Free layer of the annual Black Marble product suite (VNP46A4).

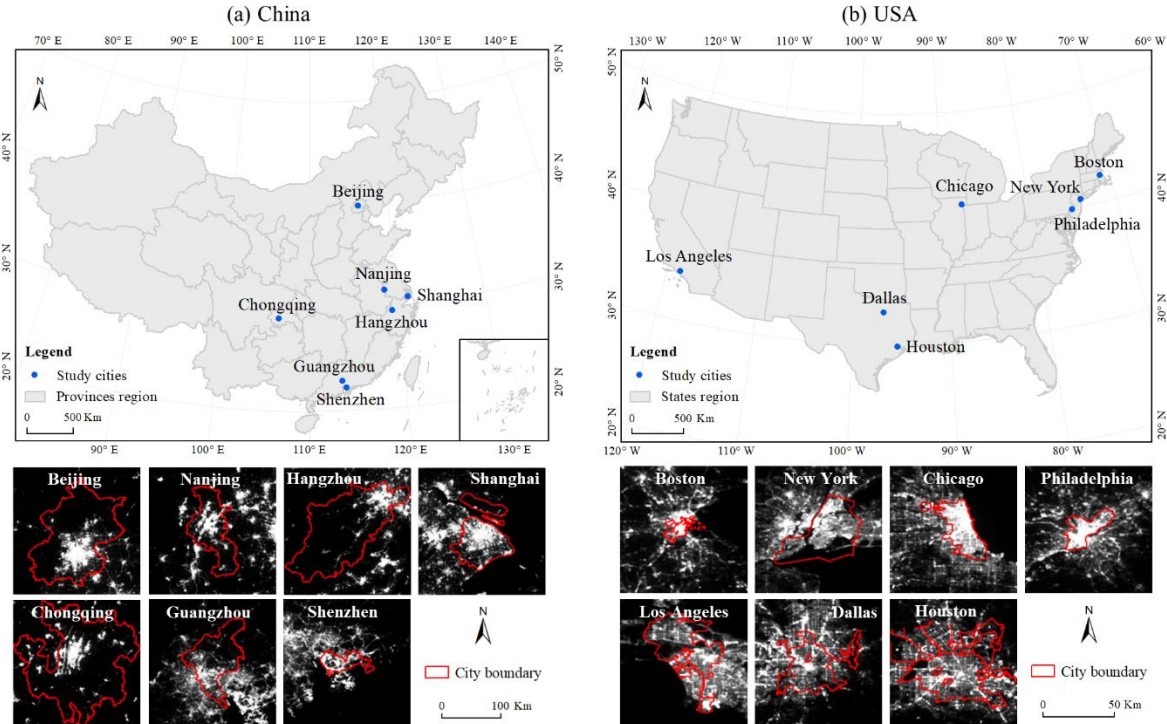

**Figure 1.** Location and NPP-VIIRS nighttime light images of all study cities. (**a**) The seven selected cities in China; (**b**) The seven selected cities in the United States.

**Table 1.** All NTL datasets used in the study.

| Dataset | Layer | Description | Unit | Purpose |
|---|---|---|---|---|
| VNP46A1 | Sensor_Zenith | Sensor zenith angle | Degrees | Fitting angular effects |
| VNP46A2 | DNB_BRDF-Corrected_NTL | BRDF corrected DNB NTL | $nW \cdot cm^{-2} \cdot sr^{-1}$ | Fitting angular effects |
| VNP46A2 | Mandatory_Quality_Flag | Mandatory quality flag | Unitless | Cloud contamination reduction |
| VNP46A2 | QF_Cloud_Mask | Quality flag for cloud mask | Unitless | Cloud contamination reduction |
| VNP46A4 | AllAngle_Composite_Snow_Free | Yearly radiance composite during snow-free period | $nW \cdot cm^{-2} \cdot sr^{-1}$ | Acquiring Potential Active Areas |

2.2.2. Auxiliary Datasets

The auxiliary data includes city boundary and reference CBDs boundary data. The city boundary data were used to clip the study area, and the city CBD boundary data were used as validation data. All CBDs are the primary CBDs of each city. Firstly, regarding the city boundary, the seven city boundaries in China are from the Resource and Environment Science and Data Center (https://www.resdc.cn/), and the seven city boundaries in the United States are from the official websites of each city. Secondly, there are some differences in the collection of urban reference CBD boundaries between China and the United States. Since the construction of urban CBD in the United States is dominated by the market economy, while the construction of urban CBD in China is dominated by the government's planning [51], that is to say, the reference CBD boundaries in China [22,23] can be determined based on planning documents or other official documents (Table 2). To get reliable reference CBD boundaries for the United States, we manually delineated CBDs from Google 3D maps based on the CBD characteristics [14,20,26]: (1) the concentration of commercial high-rise buildings; (2) enclosed by roads or riverside; (3) a clearly distinctive boundary from low buildings; (4) the urban downtown. The results of the delimitation are shown in Figure 2. It should be noted that many studies treated the area south of Manhattan's Center Garden in New York as a large CBD, but it is known from the New York Google 3D images that the building height in the area south of Center Garden shows

a high-low-high phenomenon [52]. Combining the most apparent characteristics of CBD (i.e., the concentration of high-rise density) and the special planning of New York City for Midtown Manhattan and Lower Manhattan (https://zr.planning.nyc.gov/), this paper here divides the area south of Center Garden into two important CBD areas in New York, namely Midtown Manhattan and Lower Manhattan.

**Table 2.** The reference list of CBD boundaries for seven Chinese cities.

| City | Government Documents or Official Websites |
|---|---|
| Beijing | General Office of Beijing Municipal People's Government on Accelerating the Construction of Beijing's Business Center District Interim Measures |
| Shanghai | Shanghai Pudong District Portal (https://www.pudong.gov.cn/023002007/20211215/247265.html/ (accessed on 12 October 2022)) |
| Guangzhou | Outline of the 14th Five-Year Plan and 2035 Vision for National Economic and Social Development of Tianhe District, Guangzhou |
| Shenzhen | Comprehensive Plan of Shenzhen City (2010–2020) |
| Nanjing | Nanjing Jianye High-Tech Zone Control Detailed Planning and Urban Design Integration Plan |
| Hangzhou | Hangzhou (Wulin) Central Business District "12th Five-Year Plan" Special Development Plan |
| Chongqing | Chongqing Central Business District Upgrading Action Plan (2021–2025) (Draft for Comments) |

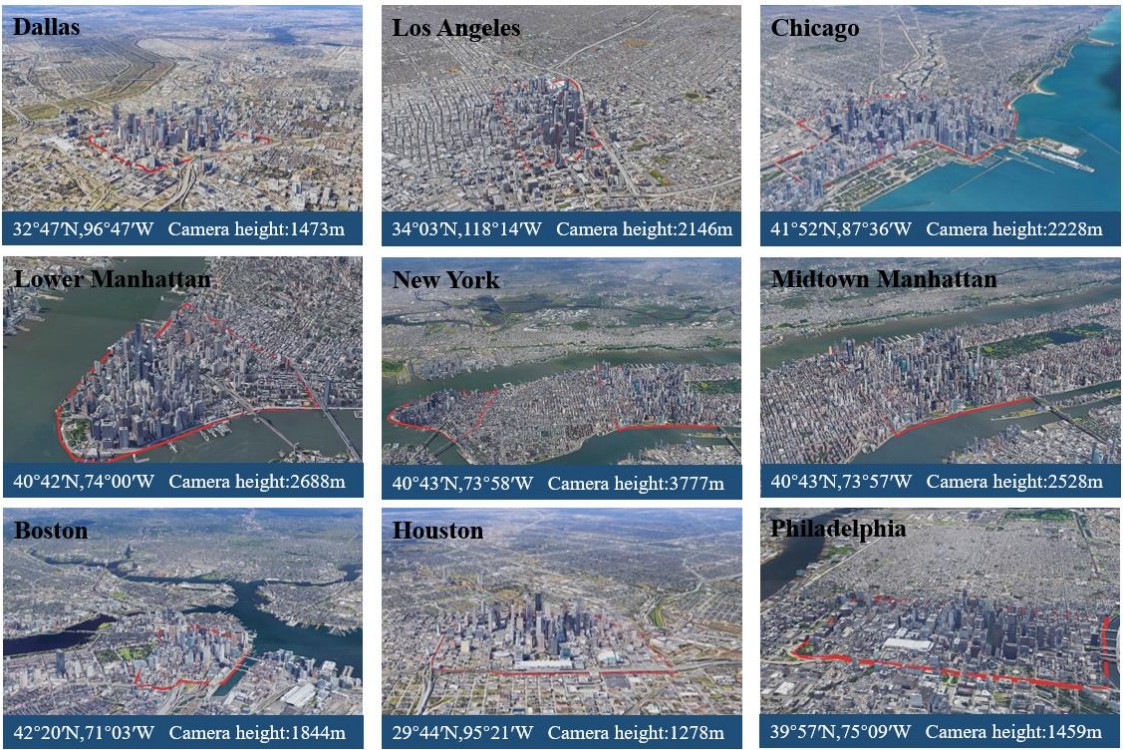

**Figure 2.** The CBD boundaries (red polygon) and Google Earth 3D images of the seven selected cities in the United States.

### 2.3. Methods

Based on NTL intensity and angular effects, we proposed a new method for CBD identification and delimitation in the study area. The workflow consists of the following steps: (a) angular effect quantification and classification; (b) using localized contour tree algorithm to acquire urban potential active area; (c) designing the rules for identifying CBD and evaluation (Figure 3).

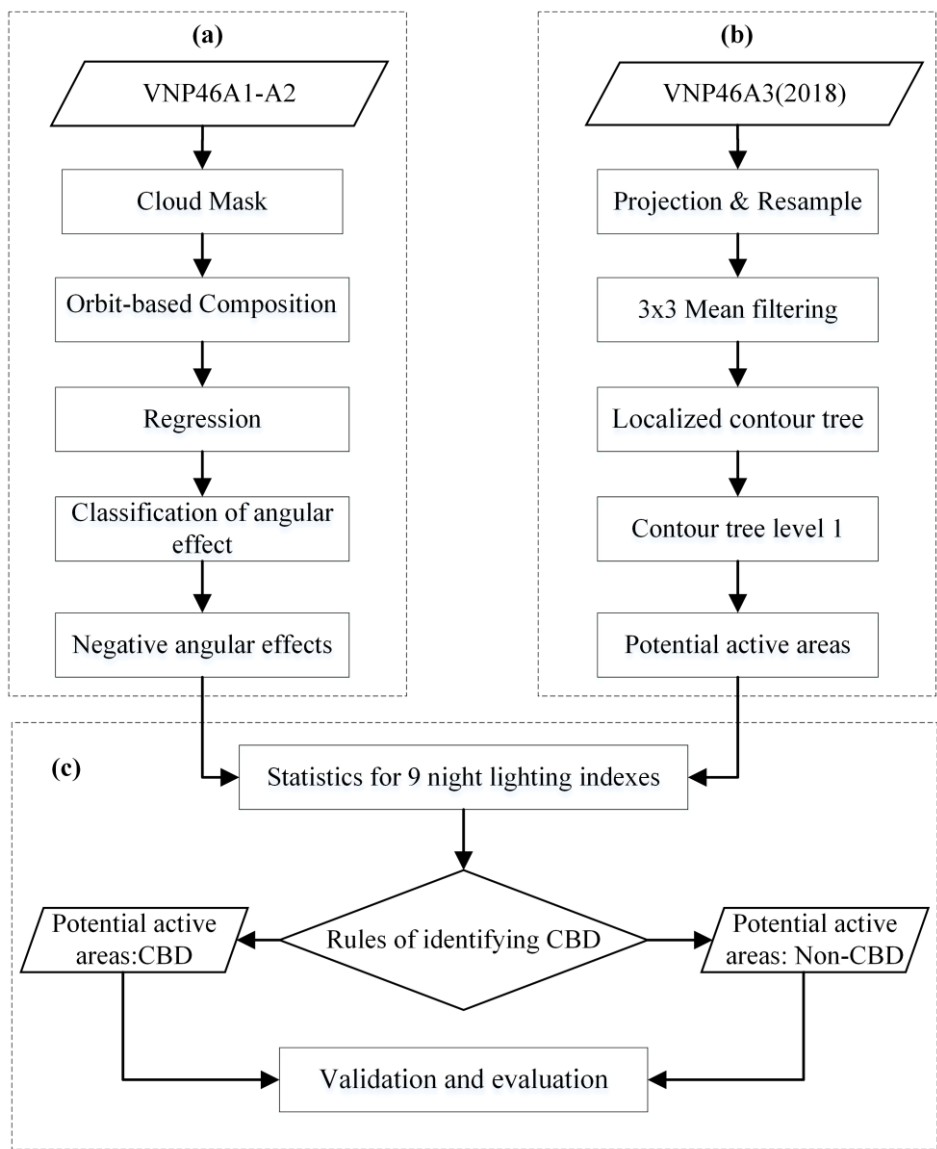

**Figure 3.** Workflow of NTL method for CBD delimitation. (**a**) Angular effect quantification and classification; (**b**) Using localized contour tree algorithm to acquire urban potential active areas; (**c**) Designing the rules for identifying CBD and evaluation.

### 2.3.1. Angular Effects

Due to the blockage of light by the building, the most important factor in shaping the angular effect is the building height, which is explicitly explained by Tan et al. In Tan's research [48], the angular effects were classified into three types in different urban landscapes: negative, positive, and U-shaped. For the negative, in high-rise building areas (i.e., CBD), NTL intensity decreases with increasing VZA. For the positive, in low-rise residential areas, NTL intensity increases with increasing VZA. For the U-shaped, in medium-rise building areas, NTL intensity decreases and then increases with increasing VZA. Therefore, the negative angular effect can reflect the high building, which is a distinctive feature recognizing CBD in heterogeneous urban landscape environments.

Three main steps were performed to get the distribution of angular effects in each study city: cloud mask, orbit-based composition, regression, and classification. Firstly, the corrected NTL layers obtained from VNP46A2 need to be masked by cloud and quality tags. Only keep the High-quality in the Mandatory_Quality_Flag layers, keep High Cloud Mask Quality, Confident Clear, and No Snow in the QF_Cloud_Mask layers. Using these

flags to perform a cloud mask that ensures the pixel-wise observations are valid. To get more valid observations, we selected two years of daily Black Marble products (2017–2018), in which the number of valid observations of Los Angeles in 2017–2018 was so low that cannot support the next regression works. Combined with the development of the urban CBD in Los Angeles in the 1990s [53], we assumed that the CBD landscape has remained unchanged. Therefore, in Los Angeles, we selected daily VNP46A1 and VNP46A2 in 2014, which is reasonable and provides a certain number of valid observations to fit the next equation. Secondly, according to the orbit law of the NPP-VIIRS satellite, the orbit of the NPP satellite visiting a fixed location on a given day will always be the same as its orbit 16 days ago. Therefore, the cloud-masked daily images were divided into 16 groups according to the orbits, and each group was composed by median to eliminate the effects of thin clouds, background noise, or random errors [48]. Finally, for the quantification and classification of angular effects, we referred to Tan's quantification approach and classification rules [48], as shown in Figure 4. In addition, Tan fitted regression on an averaging window of 3 by 3 pixels, which is too large for detecting CBD. Based on the subject of this paper, the relationship between VZA and NTL was fitted on each pixel with Equations (1) and (2) at a significance *p-level* of 0.01, and the angular effects were classified into four categories: positive, U-shaped, negative, and insignificant. We only kept the negative angle effects in this paper.

$$NTL = \beta_1\theta + \beta_0 \tag{1}$$

$$NTL = \beta_2\theta^2 + \beta_1\theta + \beta_0 \tag{2}$$

where NTL is the daily NTL radiance, $\theta$ is the VZA of the pixel, $\beta_0$ is the constant, and $\beta_1$ and $\beta_2$ are the coefficients of $\theta$ and $\theta^2$.

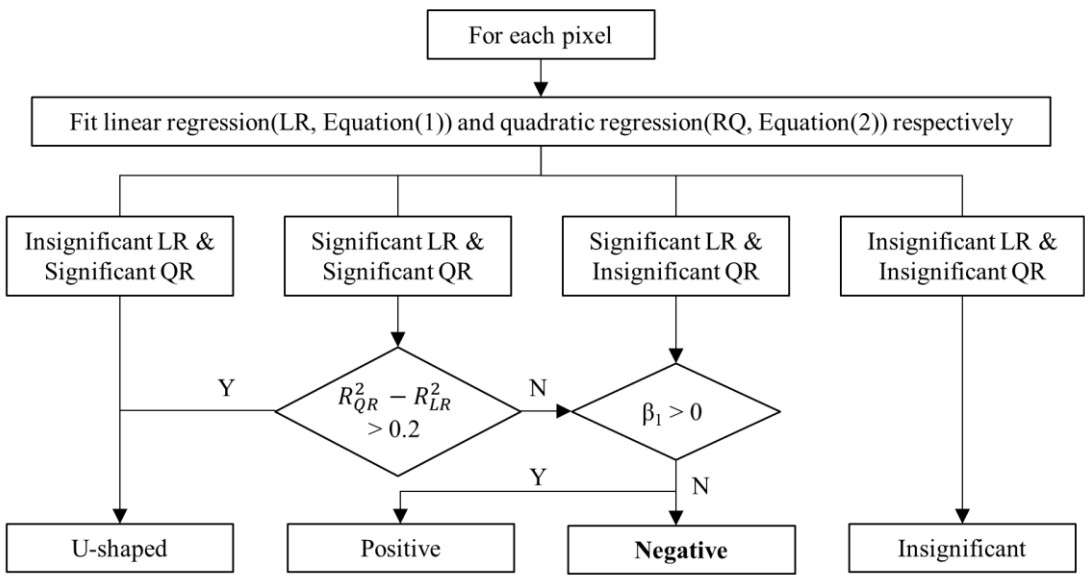

**Figure 4.** Quantification approach and classification rules.

### 2.3.2. Potential Active Areas

In previous studies of CBD delimitation, researchers depicted amounts of polygons on the map that concentrates many attributes of the CBD, whereby we also needed to depict a polygon that can represent the CBD on the NTL image. According to the previous study, there are strong artificial lights in CBD areas at night [39], which means that the boundary of the CBD can be extracted based on the intensity of nighttime lights. Recently, Chen took the NTL as topographic data and considered the city center as the peak on the NTL topographic surface, finally using the local contour tree algorithm to get elemental

urban centers that are significantly brighter than the surrounding areas [38]. The idea of this method is very consistent with the strong NTL intensity of CBD and is suitable for delineating the potential activity areas of CBD. Therefore, Chen's algorithm would be adopted in this paper to obtain potential activity areas of CBD. The potential activity areas (PAAs) were derived from the annual Black Marble product VNP46A4 in 2018. Firstly, the annual NTL data were projected to the Albers Equal Area Conic projection and resampled to 500 m spatial resolution. Secondly, the image was mean filtered by a window of $3 \times 3$, which was done to reduce data noise. Thirdly, the contour interval was set to generate contour maps for NTL. Finally, a local contour tree algorithm was used to acquire PAAs. This algorithm includes three steps: finding a seed contour line, creating a regular localized contour tree, and simplifying the contour tree (Figure 5). In the first step, the closed seed contour $S_i$ is determined based on the local maximum, and the seed contours are all taken as level-1, such as $S_1$ and $S_2$. In the second step, if the nearest contour line outside the seed contour $S_i$ contains only its seed contour, then this contour also belongs to level-1, for example, the nearest contour line outside the seed contour $S_1$ is contour $T$, and contour $T$ contains only $S_1$, so $T$ also belongs to level-1. Conversely, if the nearest contour outside the seed contour $S_i$ encloses two or more seed contours or separate branches, then the contour is Level-2, for instance, the nearest contour line outside the seed contour $S_2$ is contour $U$, if the contour $U$ encloses $S_1$ and $S_2$, then $U$ is level-2. Similarly, the contour $V$ outside $U$ also encloses two seed contours and has the same spatial inclusion connection as $U$, so $V$ is level-2. After such a local rule, all contour lines are marked out with the corresponding level and the regular contour tree is generated (Figure 5b). In the third step, we need to simplify the regular contour tree to show the level structure of the mounts. Take $T$-$S_1$ in Figure 5b, $T$ and $S_1$ are both same levels and hanged on the same branch, branching relationships like this are reduced to a single node $T$ with a large extent to represent this one branch. Similarly, the $V$ node is chosen as the representation of the level-2 structure in the $V$-$U$ branch (Figure 5c). Finally, in this paper, only the simplified level-1 areas were retained as the potential active area of CBD.

In this algorithm, three parameters need to be set for detecting PAAs, including the contour interval, NTL threshold value, and minimum area of the PAA. The sensitivity analysis for these three parameters was performed within Shanghai in Chen's study (Table 3). (1) Contours less than the NTL threshold are removed, and contours greater than or equal to the NTL threshold will be retained for detecting level-1 and level-2 by the local contour tree algorithm. This NTL threshold is not only for detecting PAA but also ensuring that PAA is within the urban region. In experiment 1, the contour interval and minimum area were both unchanged, changing different NTL thresholds for analyzing the sensitivity of the threshold. It showed that the brightness threshold had very little effect on the extraction and spatial size of the city center with high brightness because the delineation of the city center is mainly based on the topological relationship of the contour distribution, not the NTL threshold. Consequently, 36 nW·cm$^{-2}$·sr$^{-1}$ was chosen as the NTL threshold in our study due to the brightness of CBD being generally high. (2) The contour interval is used to generate the contour map. The result of experiment 2 showed that the number and area of the city center were slightly affected by different contour intervals, with the number of urban centers identified remaining almost constant when contour interval was greater than or equal to 1 nW·cm$^{-2}$·sr$^{-1}$. Therefore, 2 nW·cm$^{-2}$·sr$^{-1}$ was chosen as the contour interval in our study. (3) The area of a city center below the minimum area will be removed. The result of experiment 3 showed that the number of city centers decreased as the minimum area increased. As CBD with a small area, 1km$^2$ was chosen as the minimum area based on our own experimental subject. In summary, based on Chen's sensitivity analysis of these parameters and the properties of CBD, such as high brightness and small area, we set the NTL threshold to 36 nW·cm$^{-2}$·sr$^{-1}$, the minimum area to 1km$^2$, and the contour interval to 2 nW·cm$^{-2}$·sr$^{-1}$ in all study cities.

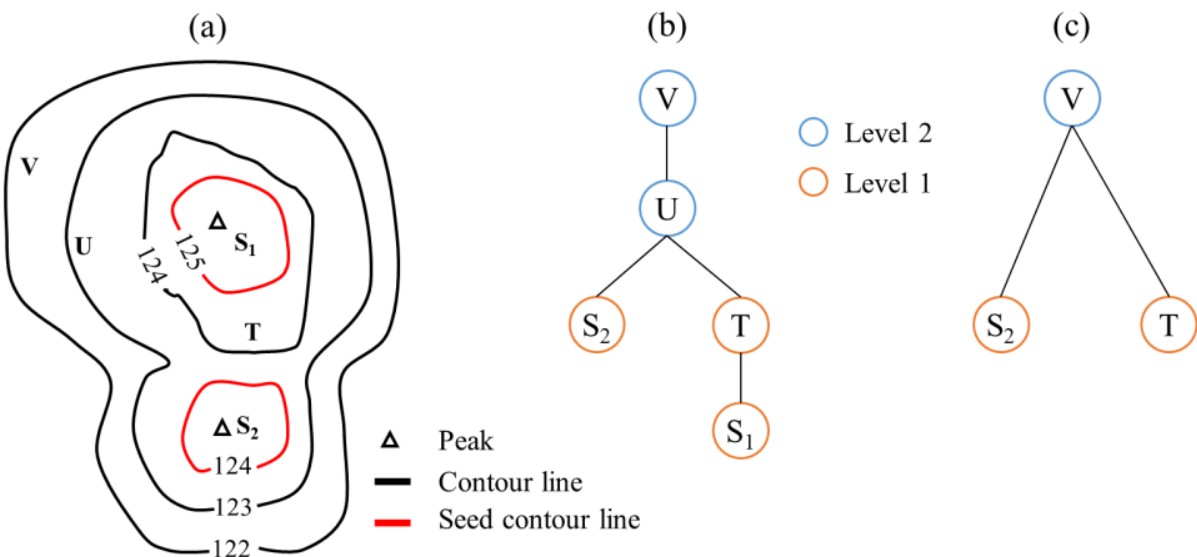

**Figure 5.** Illustration of localized contour tree algorithm. (**a**) Contour map of NTL intensity; (**b**) Regular contour tree; (**c**) Simplified level of contour tree.

**Table 3.** Parameters of three experiments in Chen's study.

| Experiment ID | NTL Threshold $(nW \cdot cm^{-2} \cdot sr^{-1})$ | Contour Interval $(nW \cdot cm^{-2} \cdot sr^{-1})$ | Minimum Area $(Km^2)$ |
|---|---|---|---|
| 1 | 30, 32, 34, 36 | 1 | 5 |
| 2 | 34 | 0.5, 1, 1.5, 2 | 5 |
| 3 | 34 | 1 | 0, 1, 2, 3, 4, 5, 6, 7 |

### 2.3.3. Rules for Identifying CBD

This section analyzed and developed the respective CBD identification rules for China and the United States separately. First, nine indexes within each PAA were counted (Table 4), where the NTL radiance values were counted based on the annual VNP46A4 in 2018. Then, according to the spatial location matches between PAAs and urban CBD boundaries in the auxiliary data, the PAAs were divided into two groups: CBD areas and non-CBD areas. Finally, box plots were used to analyze the differences in the nine indicators between CBD areas and non-CBD areas, and these differences were used to customize the CBD identification rules for China and the United States. These rules were developed by the threshold of indexes to ensure 100% producer accuracy (PA) and the highest possible user accuracy (UA). When all these rules were met, the PAA would be identified as a CBD.

**Table 4.** A specific description of the nine nighttime lighting statistics indexes.

| Index | Definition |
|---|---|
| Count_A | Number of negative angular pixels in the potential active area |
| Min_A | Minimum brightness value of negative angular pixels in the potentially active area |
| Max_A | Maximum brightness value of negative angular pixels in the potentially active area |
| Mean_A | Average brightness value of negative angular pixels in the potentially active area |
| SD_A | Standard deviation brightness value of negative angular pixels in the potentially active area |
| Mean | Average brightness values within the potential active area |
| Min | Minimum brightness values within the potential active area |
| Max | Maximum brightness values within the potential active area |
| SD | Standard deviation brightness values within the potential active area |

$$Producer's\ Accuracy = \frac{\text{Number of correctly classified PAA of a type}}{\text{Total number of PAA of that type in observation}} \tag{3}$$

$$User's\ Accuracy = \frac{\text{Number of correctly classified PAA of a type}}{\text{Total number of PAA of that type in classification}} \tag{4}$$

### 2.3.4. Validation and Evaluation

This section was divided into two main parts, namely location, validation, and boundary assessment, both of which were based on the urban CBD boundaries from the auxiliary data. Regarding location validation, it was necessary to check whether the CBD identified based on the NTL method matches the real CBD in space. We used precision and recall as the overlap of spatial location, and as long as one of these two metrics reached 0.6, the CBD was successfully identified in location. Finally, for boundary evaluation, we adopted the $F_1$-score and Jaccard index, and the higher the $F_1$-score and Jaccard index, the more consistent the spatial boundaries of the computed and comparative regions are. The expressions of precision, recall, $F_1$-score and Jaccard index, are as follows: where $a_{computed}$ is the area of the CBD based on the NTL method, $a_{comparative}$ is the area of the reference CBD in auxiliary data, $a_{overlap}$ is the overlap area of computed and comparative regions, and $a_{union}$ is the union area of computed and comparative regions.

$$precision = \frac{a_{overlap}}{a_{computed}} \tag{5}$$

$$precision = \frac{a_{overlap}}{a_{computed}} \tag{6}$$

$$F_1 - score = 2 * \frac{\text{precision} * \text{recall}}{\text{precision} + \text{recall}} \tag{7}$$

$$\text{Jaccard index} = \frac{a_{overlap}}{a_{union}} \tag{8}$$

## 3. Results

### 3.1. PAA and Angular Effects

Based on annual Black Marble data, multiple PAAs were delineated using the localized contour tree method for each study city (Figure 6a,c). The total number of PAAs was 198 and 185 in Chan and the U.S., respectively. These urban PAAs contained different types of urban landscapes, such as CBDs, commercial districts, airports, and factories. In terms of angular effects, there were negative angular effects inside the CBD boundaries in all study cities, especially Philadelphia and New York, where there were several negative angle effect pixels (Figure 6b,d). In addition, there were also negative angular effects in non-CBD areas, such as Beijing, Chongqing, and Guangzhou in China, where there were also negative angular effects outside CBD areas. Overall, it is difficult to identify CBD by PAA or negative angular effects alone, and the information of both should be combined to identify the CBD of each city from multiple PAAs.

### 3.2. Rules for Identifying CBD in the U.S. and China

Because we thought that the PAA should enclose the negative angular effects reflecting high-rise buildings, and for better distinguishing the difference between CBD and Non-CBD, PAAs without negative angular effects were eliminated, and the remaining PAAs were used for analyzing the difference between CBD and Non-CBD. The boxplots were made with 74 PAAs in China (CBD:7, Non-CBD:67) and 29 PAAs in the USA (CBD:8, Non-CBD:21).

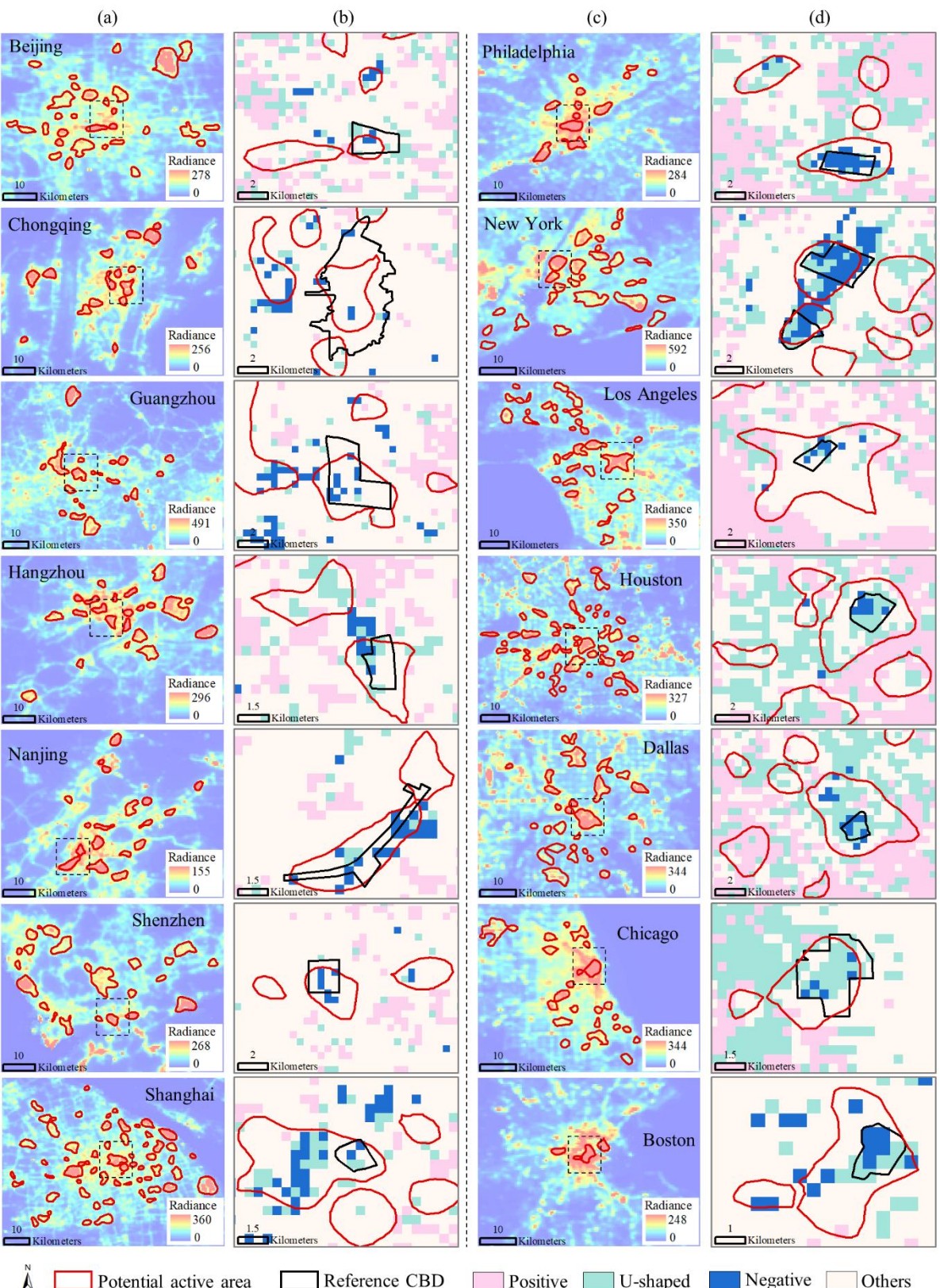

**Figure 6.** Distribution of PAAs and angular effects, where (**b**) and (**d**) are enlarged images of the black dashed boxes in (**a**) and (**c**), respectively. (**a**) Radiance distribution of Chinese cities and PAAs; (**b**) Distribution of angular effect near CBD of Chinese cities; (**c**) Radiance distribution of the U.S. cities and PAAs; (**d**) Distribution of angular effect near CBD of the U.S. cities.

As shown in Figure 7a, there were differences between CBD and Non-CBD at the median level of all indicators in China, and the median of CBD was larger than the median of non-CBD. The interquartile ranges of all indexes slightly overlap between CBD and Non-CBD, especially Count_A, Max_A, SD_A, Mean, and Min, which exhibited better separability between CBD and Non-CBD. As shown in Figure 7b, there were distinctive differences between CBD and Non-CBD at interquartile ranges of all indexes in the U.S., including the Count_A, Max_A, and Mean, which presented stronger separability between CBD and Non-CBD than China. In short, based on the comparison of box plots, several indexes were capable of distinguishing between CBD and non-CBD (i.e., Count_A and Mean), which corresponded with our assumption that the negative angular effects and NTL intensity are able to identify CBDs. Consequently, we set thresholds for these indexes to guarantee that the PA accuracy of CBD was 100% while UA was as high as possible. After several attempts, it was required 5 rules and 2 rules to satisfy our purpose of accuracy in China and the USA, respectively. The rules of China are (1) Count_A > 2; (2) Mean > 46 $nW \cdot cm^{-2} \cdot sr^{-1}$; (3) Max_A > 71 $nW \cdot cm^{-2} \cdot sr^{-1}$; (4) SD < 20 $nW \cdot cm^{-2} \cdot sr^{-1}$; and (5) Min_A > 38 $nW \cdot cm^{-2} \cdot sr^{-1}$. The rules of the USA are: (1) Count_A > 3; and (2) Mean > 120 $nW \cdot cm^{-2} \cdot sr^{-1}$. The confusion matrixes and classification accuracy indices of these rules are shown in Table 5. The China rules reached 100% for PA and 70% for UA, and all 7 urban PAAs were correctly identified as CBD, but 3 Non-CBD of PAAs were also identified as CBD. The USA rules reached 100% for both PA and UA, and it correctly distinguished 195 PAAs as CBD and Non-CBD, showing no commissions and omissions.

**Table 5.** Accuracy assessment of China rules and the USA rules. Expressed in the number of PAAs.

| Observation | Classification | | | | | |
| --- | --- | --- | --- | --- | --- | --- |
| | China | | | USA | | |
| | **CBD** | **Non-CBD** | **PA (%)** | **CBD** | **Non-CBD** | **PA (%)** |
| CBD | 7 | 0 | 100 | 8 | 0 | 100 |
| Non-CBD | 3 | 195 | 99.5 | 0 | 187 | 100 |
| UA (%) | 70 | 100 | | 100 | 100 | |

### 3.3. Results of CBDs Identification and Evaluation

As shown in Figure 8, the yellow polygons were produced by the localized contour tree method and our rules, the red polygon were urban CBD boundaries in the auxiliary data. The shapes of computed CBD were determined by the contours of its NTL intensity surface. As can be seen from Figure 8, all computed CBDs captured the corresponding positions of the reference CBD. And through the indices of location validation (Table 6), the precision or recall of every study city was more than 0.6, which indicated that our method could validly recognize and delimitate CBDs in space. Regarding the evaluation of boundary, it can be seen that $F_1$-score of several cities near or greater than 0.5, which displayed a good agreement between reference CBD and computed boundaries. Few cities' $F_1$-score were below 0.5(i.e., Shanghai, Boston, Los Angeles, Houston, and Dallas), and relevant Jaccard indexes are small because these computed CBDs all completely enclose the reference CBD in space.

**Table 6.** Evaluation of urban CBD boundaries in China and the USA.

| Country | City | Computed Area ($Km^2$) | Reference CBD Area ($Km^2$) | Precision | Recall | F1-Score | Jaccard Index |
| --- | --- | --- | --- | --- | --- | --- | --- |
| | Beijing | 2.18 | 3.83 | 0.76 | 0.43 | 0.55 | 0.38 |
| China | Chongqing | 10.79 | 25.27 | 0.93 | 0.40 | 0.56 | 0.39 |
| | Guangzhou | 15.66 | 11.36 | 0.57 | 0.79 | 0.66 | 0.49 |

| Country | City | Computed Area (Km²) | Reference CBD Area (Km²) | Precision | Recall | F1-Score | Jaccard Index |
|---|---|---|---|---|---|---|---|
| China | Hangzhou | 6.88 | 2.55 | 0.34 | 0.93 | 0.50 | 0.34 |
| | Nanjing | 10.51 | 4.10 | 0.32 | 0.81 | 0.46 | 0.30 |
| | Shanghai | 18.07 | 1.72 | 0.09 | 0.99 | 0.17 | 0.09 |
| | Shenzhen | 7.64 | 4.05 | 0.39 | 0.73 | 0.51 | 0.34 |
| USA | Boston | 8.00 | 1.57 | 0.20 | 1.00 | 0.33 | 0.20 |
| | Chicago | 10.10 | 7.39 | 0.60 | 0.82 | 0.69 | 0.53 |
| | Dallas | 21.00 | 1.79 | 0.09 | 1.00 | 0.16 | 0.09 |
| | Houston | 16.23 | 3.24 | 0.20 | 1.00 | 0.33 | 0.20 |
| | Los Angeles | 31.17 | 2.22 | 0.07 | 1.00 | 0.13 | 0.07 |
| | New York Lower Manhattan | 5.10 | 3.46 | 0.57 | 0.83 | 0.67 | 0.51 |
| | New York Midtown Manhattan | 10.62 | 6.52 | 0.51 | 0.83 | 0.63 | 0.46 |
| | Philadelphia | 9.97 | 3.15 | 0.32 | 1.00 | 0.48 | 0.32 |

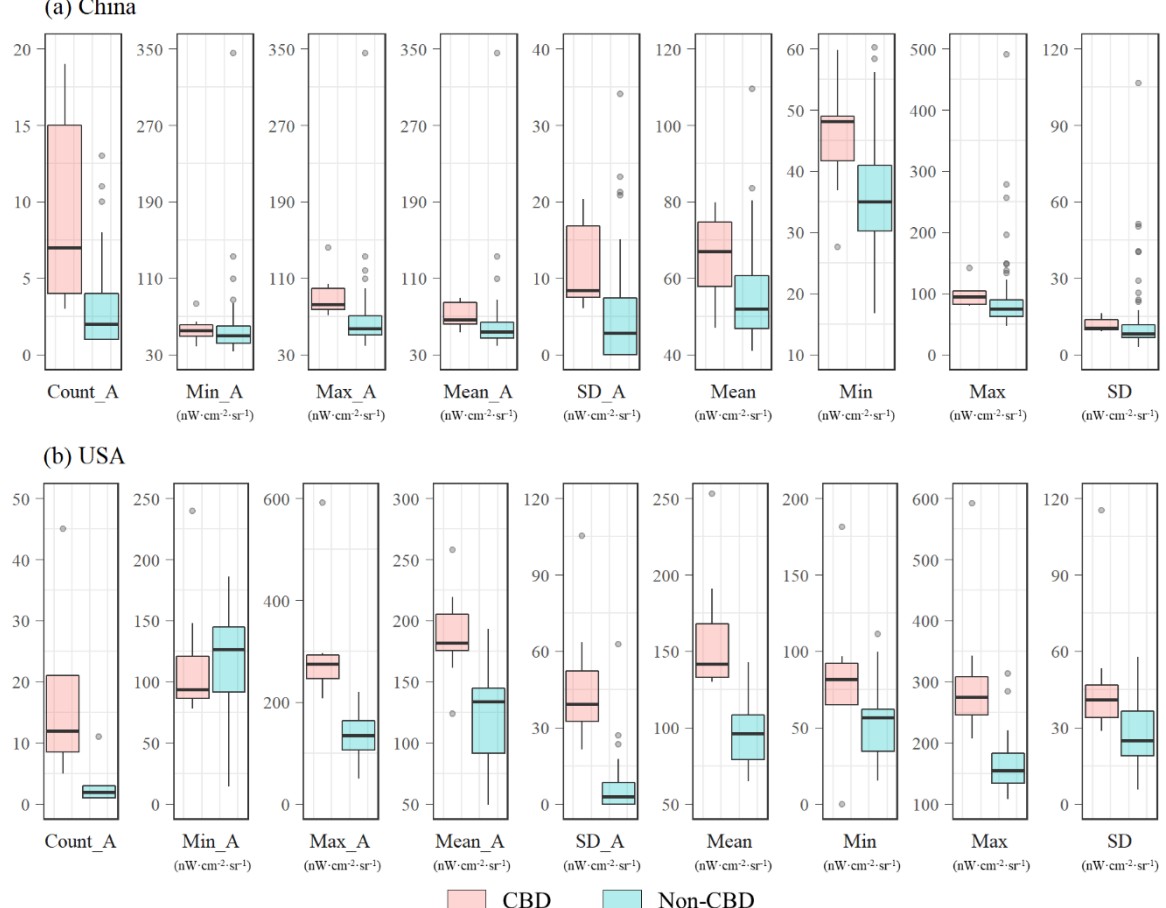

**Figure 7.** The boxplots of nine indexes are based on PAAs of CBD and Non-CBD. (**a,b**) The boxplots of China and the USA.

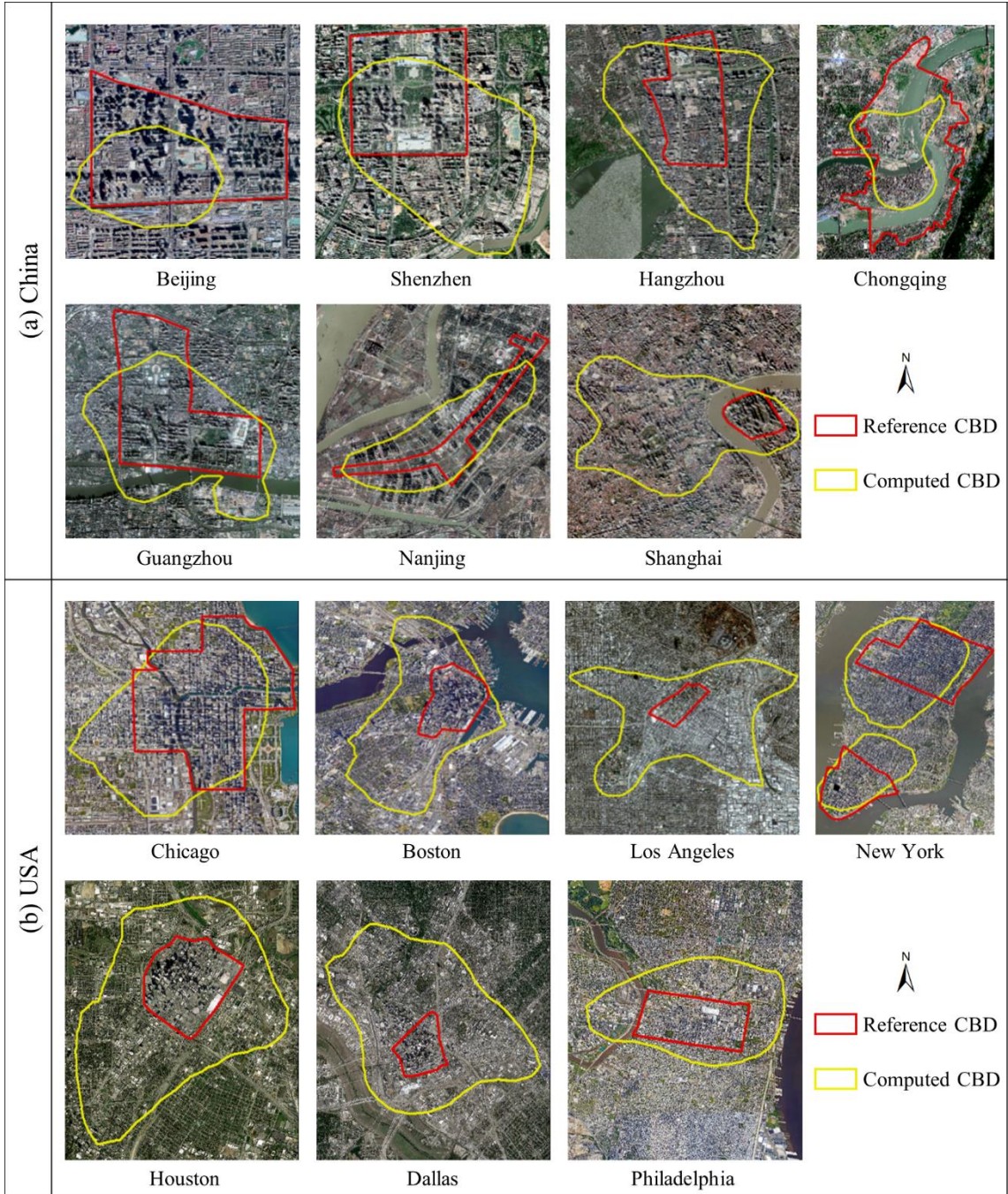

**Figure 8.** The reference and computed CBD boundaries and Google Earth image of the study cities.

## 4. Discussion

### 4.1. Advantages of the New Method

The CBD detection and extraction method proposed in this paper has three advantages. First, our method can be applied globally thanks to angular effects and Black Marble products. Black Marble products are a set of free, long-time series and high-quality global NTL remote sensing images [32]. In our technical process, we relied on the annual and daily data of Black Marble products to obtain the potential activity areas and angular effects of the city, to achieve CBD detection and delineation. The most critical part of this is the role of negative angular effects. If the CBDs are extracted by the NTL brightness alone, then only the spatial distribution structure of the city can be obtained. For example, the

localized contour tree method [38] and city center development index [54] are capable of generating the urban potential activity areas, which are CBD, airports, industry, etc. Even among the previous GIS studies [22,24], multiple hotspot areas were generated by POI data only, which are mainly large and small commercial areas. However, there is usually only one primary CBD in a city. It is very inefficient to rely on local knowledge to identify CBDs on a large scale, as previous authors have done. Therefore, it must be combined with the unique features of CBDs for automated identification. Combining the negative angular effects and the NTL intensity can globally identify urban CBDs. This is because the angular effects have a relationship with building height, and the intensity of commercial activities in CBDs is positively related to the NTL intensity. These two strong relationships help us to identify CBDs from many potential activity areas; even the nighttime light radiance of CBDs will increase in the future The USA's CBD is an example of exactly this. We can see from the box plot (Figure 7), the PAA of the Chinese CBD is between 40–80 nW·cm$^{-2}$·sr$^{-1}$, and the PAA of the US CBD is between 125–200 nW·cm$^{-2}$·sr$^{-1}$ on the ***Mean***. This huge difference is caused by the fact that the USA's CBDs are stronger than the Chinese's CBDs. Therefore, the rules of identifying CBD between China and USA are different. It also indicates that when the brightness of the Chinese CBD reaches the current level of the USA, a stable angular effect can still be extracted. Second, our method can offer CBDs with a uniform criterion. From Table 6 and Figure 8, it can be seen that the reference CBD and the computed boundary are generally consistent in space. Since the reference CBD in China is obtained based on planning documents, the reference CBD in the U.S. is obtained based on high-rise clusters, and the computed boundary only relies on the spatial distribution characteristics of NTL. However, this does not mean that the computed boundary loses its value. Urban planning is customized by planners, and different countries and cities have different customization rules that produce different CBDs at variance criteria. When it is necessary to compare the information within each CBD, the requirement for the boundary is the CBD under the unified standard. And our approach just can get the CBD boundary under the unified standard in an objective and quantitative way, which is based on a set of global data with the same parameters for boundary extraction in each city. Each PAA is delimited based on the natural density and intensity of the city's NTL distribution, which means CBDs are unified under the standard of NTL. Thirdly, the CBD delineated by our proposed method is more consistent with the reference CBD than other methods. We collected the map of CBD extraction results from previous studies, whose study city is common to this research. These maps were from the study of Yang [24], Yu [23], and Wu [22] for the cities of Shenzhen, Guangzhou, and Nanjing. These methods used POI data or a combination of POI and road nodes. As shown in Figure 9 below, the areas extracted by these GIS-type methods are more like functional core areas located in or near the CBD. Due to the fine scale of POI data and its wealth of information, some methods can detect multiple hotspot areas, such as Planar KDE and Network KDE for Guangzhou, which extract two regions. Our method extracts a larger area of CBD coverage that matches the reference CBD. Therefore, when delineating the CBD of a city, we recommend using our proposed method. When studying the functional core of a city, we recommend using the POI-based KDE method.

*4.2. Limitations*

The approach we proposed still has a few limitations. In terms of the number of rules in both countries, the U.S. rules are simpler and more effective than the China rules. In identifying CBD, the USA only needed to use Count_A and Mean, which are two conditions to completely separate CBD from Non-CBD. In contrast, China needed 5 conditions such as Count_A, Mean, SD, Min_A, Max_A to identify CBD, and even with the combination of 5 rules, 3 Non-CBDs are incorrectly identified as CBDs. If China relies only on Count_A and Mean, as the USA did, then within the set of rules with Count_A >2 and Mean >46 nW·cm$^{-2}$·sr$^{-1}$, 17 Non-CBDs are misclassified as CBD. The reason for this is that Chinese cities have formed many commercial districts and high-rise neighborhoods

with nighttime lighting and building heights similar to the local CBD. The 3 non-CBDs in the five rules are the Yansha commercial district in Beijing and two high-rise residential districts in Nanjing. Similarly, in Taubenböck's identification [26], the high-rise residential areas in Paris were also identified as CBDs. Therefore, high-rise residential areas are usually mixed with CBDs in the results. Apart from this, the U.S. rules perform well because CBDs are significantly brighter than Non-CBDs in U.S. cities (Figure 7b), and the distribution of building heights in U.S. cities is not as complex as in China, where building heights follow an almost decreasing pattern from the city center [48], i.e., the building height of multiple Non-CBDs are low. Therefore, the rules need to be determined by the characteristics of each country's urban landscape and economic development. The brightness of CBD and spatial distribution of urban building heights are essential factors influencing the variation with the number and threshold of rules. Consequently, to quantitatively analyze the number of rules and the thresholds of indexes, applying more cities and periods is necessary for future study.

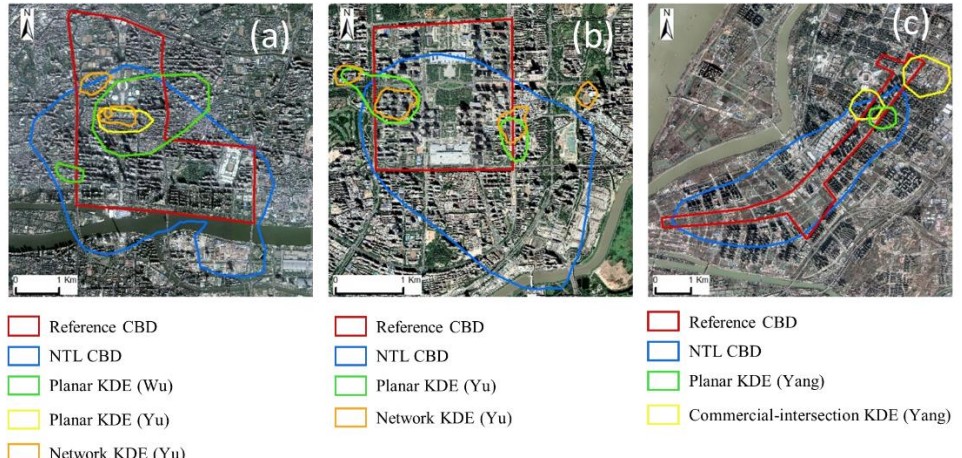

**Figure 9.** The comparison of CBDs from the proposed new method (NTL CBD) and GIS-based methods. (**a**) Guangzhou; (**b**) Shenzhen; (**c**) Nanjing.

*4.3. Recommendations for Future Studies*

There are three recommendations for future studies. Firstly, before applying our method, we suggest that researchers should pay attention to whether the number of valid observations in the study area is sufficient to establish the regression relationship between VZA and NTL when obtaining the angular effects. If the number of valid observations is not sufficient, the regression relationship will show insignificance, and such a result will mislead us to believe that there is no angular effect in the region. Secondly, when applying the method to other cities, attention should be paid to the landscape characteristics of the local cities and the size of the CBD. For example, traditional cities like Lucca in Italy and Vienna in Australia are not suitable because they don't have the characteristics of a modern CBD [55]. It is feasible to target modern cities like China and the U.S., such as Toronto in Canada. And some CBDs might be smaller than one pixel (0.25 km$^2$), in which case it's not appropriate to detect by our method. Our approach is applicable to global megacities, where the CBD is typically larger than 0.25 km$^2$. This paper explored the feasibility of detecting CBD using NTL remote sensing and hoped that higher spatial resolution NTL data will be available to support the fitting of angular effects for smaller CBD detection in the future. Finally, the approach of this paper can help identify urban centers. Although in terms of definition and spatial extent, urban centers are broader than CBDs. For example, the urban center can be the economic, cultural or political center, while a CBD is just an economic center [56]. From this perspective, the negative angular effects and NTL brightness can provide features for identifying not only CBDs but perhaps those urban economic centers as well.

## 5. Conclusions

CBD is the most important type of urban structure and the core of a city. In this paper, we proposed a new method for identifying and delineating CBDs based on NTL, which had been successfully implemented in seven major cities in two countries, China and the United States, respectively. The unification of data and contours is key to the standardization of delineated boundaries. Thanks to a set of global Black Marble products and natural surfaces such as NTL contours, we identified the primary CBDs that were delineated on a framework based on a unified standard, which helps a lot in urban geographic analysis between CBDs of different cities. It was possible to perform semi-automatic identification because we combined more nighttime lighting characteristics of CBDs, including brightness and negative angular effects that can reflect high-rise buildings, while previous studies only resorted to the density of commercial activities or the feature of the building height. In summary, the method in this paper enables large-scale CBD identification and delimitation and helps researchers to perform comparative analysis and urban centrality calculations in urban geography. It provides a new approach to delineating CBD for cities lacking regional data and exploits the potential of nighttime light remote sensing.

**Author Contributions:** Conceptualization, X.C. (Xin Cao) and J.C.; methodology, N.J. and X.C. (Xuehong Chen); software, N.J.; validation, N.J.; formal analysis, N.J. and X.C. (Xin Cao); investigation, X.C. (Xin Cao), N.J. and J.C.; resources, X.C. (Xin Cao), J.C. and X.C. (Xuehong Chen); data curation, N.J. and X.C. (Xuehong Chen); writing—original draft preparation, N.J.; writing—review and editing, X.C. (Xin Cao) and X.C. (Xuehong Chen); supervision, X.C. (Xin Cao) and J.C.; project administration, X.C. (Xin Cao); funding acquisition, X.C. (Xin Cao). All authors have read and agreed to the published version of the manuscript.

**Funding:** This research was funded by the Special Project of Science and Technology Basic Resources Survey, China Ministry of Science and Technology under Grant 2019FY202502.

**Data Availability Statement:** Daily Black Marble images (VNP46A1-A2) and annual Black Marble images (VNP46A4) are available at NASA website (https://blackmarble.gsfc.nasa.gov/ (accessed on 11 October 2022)). The seven city boundaries in China are provided by China Resource and Environment Science and Data Center (https://www.resdc.cn (accessed on 11 October 2022)). The seven city boundaries in the United States are from the official websites of each city (Boston, https://data.boston.gov/dataset/city-of-boston-boundary (accessed on 11 October 2022); New York, http://gis.ny.gov/gisdata/inventories/details.cfm?DSID=927 (accessed on 11 October 2022); Chicago, https://data.cityofchicago.org/Facilities-Geographic-Boundaries/Boundaries-City/ewy2-6yfk (accessed on 11 October 2022); Philadelphia, https://www.opendataphilly.org/dataset/city-limits (accessed on 11 October 2022); Los Angeles, https://geohub.lacity.org/datasets/lahub::city-boundary/explore?location=34.019779%2C-118.412043%2C10.53 (accessed on 11 October 2022); Dallas, https://www.dallasopendata.com/Services/FY-2017-City-of-Dallas-City-Limits/ad4m-4kje (accessed on 11 October 2022); Houston, https://mycity.houstontx.gov/houstonmapviewer/ (accessed on 11 October 2022)).

**Conflicts of Interest:** The authors declare no conflict of interest.

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
