# Peer review of "A New Method for Identifying the Central Business Districts with Nighttime Light Radiance and Angular Effects"

_remotesensing, doi:10.3390/rs15010239_

Round 1

Reviewer 1 Report

The paper is ready for publication.

Author Response

Dear reviewer,

Thank you for your nice comments on our article. We are very grateful for your support.

Reviewer 2 Report

This manuscript proposes a new method to identify the central business districts using Black Marble products. I think the manuscript is well organized to prove the feasibility of the novel method. I recommend this paper to be published after considering the following points:

1)     Some formatting issues need to be noticed. In Figure 2, is the CBD boundaries and Google Earth 3D images of Philadelphia missing? In addition, some spelling errors in the text should be corrected and the language needs to be polished.

2)     In Section 2.2.1, I suggest that the authors describe the various Black Marble layers with a table, which contains the layer's name, description, parameters, purpose and other attributes. In Figure 6, I suggest to superimpose PAA in the enlarged map ((b) and (d)), in order to see the positional relationship between PAA and the reference CBD more clearly.

3)     In Section 3.2, when calculating the confusion matrix and classification accuracy metrics, what is the criteria to tell if a PAA is correctly identified as CBD?

4)     In this study, linear regression and quadratic regression were used for each pixel. Could the possible geometric bias between images of VNP46A2 have an impact on the significance p-level? Why not perform regression based on a window centered around pixels?

5)     Chinese cities are developing rapidly, the nighttime light will change over time even if the area is observed from the same angle. How does this interfere with the angle effect of nighttime light radiance? What impact will this have on identifying CBD

6)     In Section 4.2, “Therefore, the rules need to be determined by the characteristics of each country's urban landscape and economic development.”. Can the authors describe in a quantitative way how the number of rules and the thresholds of indexes are formulated? Besides, I suggest that the author expand the study area to prove the universality of the method.

7)     In this study, the positions and boundaries of CBD depend on PAA, so the thresholds setting during the generation of PAA is very important. How to prove that the selected thresholds are the most appropriate for generating the PAA? What impact will different thresholds have on identifying CBD?

8)     In this study, only location verification and boundary assessment were used to verify the method. I suggest that the author compare the proposed method with other CBD identification methods and develop a more objective and detailed evaluation.

9) Important literature missing:

     Li, X., Shang, X., Zhang, Q., Li, D., Chen, F., Jia, M., Wang, Y., 2022. Using radiant intensity to characterize the anisotropy of satellite-derived city light at night. Remote Sensing of Environment. 271, 112920

     Tong, K.P., Kyba, C.C.M., Heygster, G., Kuechly, H.U., Notholt, J., Kollth, Z., 2020. Angular distribution of upwelling artificial light in Europe as observed by Suomi–NPP satellite. Journal of Quantitative Spectroscopy and Radiative Transfer, 107009

     Kyba, C., Ruhtz, T., Lindemann, C., Fischer, J., Hölker, F., 2013. Two camera system for measurement of urban uplight angular distribution. In: International Radiation Symposium (IRC/IAMAS) Radiation Processes in the Atmosphere and Ocean.

Author Response

Dear reviewer,

We feel great thanks for your professional review work on our article. As you are concerned, there are several problems that need to be addressed. According to your nice suggestions, we have made extensive corrections to our previous draft. Appended to this letter is our point-by-point response to the comments raised by the reviewers. The comments are reproduced and our responses are given directly afterward in a different color (red).

Reviewer 3 Report

The article seems interesting to me.

Some suggestions:

1) Whether the spatial resolution of VIIRS-produced data is appropriate to discover CBDs? In some cities, CBD might be smaller than even one pixel (~500 m2). Please add this point to the discussion section. Please specify which was the resolution of NTL images in the work of Tan et al. which you cite.

2) The authors argue that their rules might be applied globally to define CBDs. In the meantime, the rules are different for CBDs in China and in the US. So, which rules to apply to define CBDs, say, in Europe?

3) Could the authors add cross-validation results - that is, applying Chinese rules to CBDs in the US and vice versa? At that, report those cross-validation results in a form similar to those reported in Tables 3 and 4.

4) Please also add to Table 4 the Jaccard index (calculated as a_overlap/(a_computed + a_comparative)).

5) In the discussion section, a comparison of the obtained precision levels with the ones of previous studies on the topic is required.

6) Please describe in short the work of Chen on the sensitivity analysis: which areas were analyzed there? Whether a single NTL threshold of 36 nW/cm2/sr is appropriate for both China and the US? 

Author Response

Dear reviewer,

Thank you again for your positive comments and valuable suggestions to improve the quality of our manuscript. Please see the attachment. Appended to this letter is our point-by-point response to the comments raised by the reviewers. The comments are reproduced and our responses are given directly afterward in a different color (red).

Round 2

Reviewer 2 Report

The manuscript has been well revised according to  my comments.